# A cautionary note on the use of N-acetylcysteine as a reactive oxygen species antagonist to assess copper mediated cell death

**Rebecca E. Graham**[1], **Richard J. R. Elliott**[2], **Alison F. Munro**[2], **Neil O. Carragher**[2]*

**1** Centre for Clinical Brain Sciences, University of Edinburgh, Edinburgh, United Kingdom, **2** Cancer Research UK Scotland Centre, Institute of Genetics and Cancer, University of Edinburgh, Western General Hospital, Edinburgh, United Kingdom

* N.Carragher@ed.ac.uk

## Abstract

A new form of cell death has recently been proposed involving copper-induced cell death, termed cuproptosis. This new form of cell death has been widely studied in relation to a novel class of copper ionophores, including elesclomol and disulfiram. However, the exact mechanism leading to cell death remains contentious. The oldest and most widely accepted biological mechanism is that the accumulated intracellular copper leads to excessive build-up of reactive oxygen species and that this is what ultimately leads to cell death. Most of this evidence is largely based on studies using N-acetylcysteine (NAC), an antioxidant, to relieve the oxidative stress and prevent cell death. However, here we have demonstrated using inductively coupled mass-spectrometry, that NAC pretreatment significantly reduces intra-cellular copper uptake triggered by the ionophores, elesclomol and disulfiram, suggesting that reduction in copper uptake, rather than the antioxidant activity of NAC, is responsible for the diminished cell death. We present further data showing that key mediators of reactive oxygen species are not upregulated in response to elesclomol treatment, and further that sensitivity of cancer cell lines to reactive oxygen species does not correlate with sensitivity to these copper ionophores. Our findings are in line with several recent studies proposing the mechanism of cuproptosis is instead via copper mediated aggregation of proteins, resulting in proteotoxic stress leading to cell death. Overall, it is vital to disseminate this key piece of information regarding NAC's activity on copper uptake since new research attributing the effect of NAC on copper ionophore activity to quenching of reactive oxygen species is being published regularly and our studies suggest their conclusions may be misleading.

## Introduction

It has recently been revealed that a novel class of copper ionophores display highly selective and potent cytotoxic activity towards cancer cells. These compounds bind free copper in the

**Data Availability Statement:** All relevant data are within the paper and its Supporting Information files.

**Funding:** This study was supported by a Cancer Research UK award (CRUK Small Molecule Drug Discovery project) awarded to NOC (C42454/A24892) and an MRC funded 4-year multi-disciplinary PhD studentship offered by the University of Edinburgh, awarded to REG. NOC also received support from Anne Forrest Fund for Oesophageal Cancer Research at the University of Edinburgh, under grant number D61720 to support this study. The funders had no role in study design, data collection and analysis, decision to publish, or preparation of the manuscript.

**Competing interests:** The authors have declared that no competing interests exist.

media and act as a shuttle, bringing it into cancer cells selectively where it accumulates and induces cytotoxic activity [1–4].

Structurally this class of copper ionophores, including elesclomol and disulfiram contain thiocarbonyl groups, and they are known copper chelators. Disulfiram, a thiuram disulphide, breaks down in acidic or Cu(II) rich environments to produce a dithiocarbamate; diethyl-dithiocarbamate (DDTC) [5]. DDTC, like other dithiocarbamates, are known to form complexes with transition elements, but are most stable in a Cu(II) chelate [6]. Although elesclomol is structurally unrelated to the dithiocarbamates, it also forms organometallic complexes, particularly with Cu(II), due to its two thiocarbonyl moieties [7, 8]. We and others, have previously demonstrated that elesclomol and disulfiram act as copper ionophores [1–4, 9]. Incubation of cancer cells with elesclomol and disulfiram leads to a significant increase in intracellular copper levels and eventually cell death. Critically, removal of free copper from the media prior to drug treatment results in complete loss of the drugs cytotoxic effects, demonstrating the essentiality of copper in the cytotoxic mechanism-of-action of these compounds [1–4, 9].

While there is now much evidence demonstrating the role of intracellular copper accumulation in the mechanism of these compounds, the mechanism leading to the cell death downstream of the copper accumulation is still contested. Proposed mechanisms include the production of reactive oxygen species (ROS) [4, 10–13], inhibition of the ubiquitin-proteasome system [14–16], and interference with the mitochondrial electron transport chain [17, 18]. While the mechanism remains to be fully elucidated or may involve multiple mechanisms, the most widely published mechanism involves ROS production [4, 10–13]. Since copper is a redox active metal, it has been proposed that the intracellular copper accumulation leads to the production of ROS and that this is the cause of cell death [4, 10–13]. Much of the evidence for this is based on the ability of N-acetylcysteine (NAC) to alleviate the toxicity of the drugs [10, 12, 13, 19]. NAC is a precursor of L-cysteine that results in glutathione biosynthesis [20]. Further NAC is widely used *in vivo* and *in vitro as a universal cytoprotective antioxidant* [21, 22]. Due to the fact that NAC is widely reported in the literature to be a ROS antagonist, the alleviation of elesclomol and other copper ionophore induced cell death by NAC has been interpreted to support the claim that ROS may underlie the mechanism by which copper ionophores induce cuproptosis. However, the effects of NAC on ROS may be misreported and misunderstood [20, 23] and further NAC has other mechanisms, including an ability to form conjugates with copper [24], thiol-reactive compounds [25] and other electrophiles [20]. Here we explore the mechanism through which NAC mediates relief of copper ionophore induced cell death and importantly demonstrate that it is not via the widely accepted effects of ROS antagonism (Fig 1).

## Materials and methods

### Cell culture

Oesophageal adenocarcinoma (OAC) lines were grown in Roswell Park Memorial Institute (RPMI) (Life Technologies; #11875101) supplemented with FBS (10%, Life Technologies; #16140071) and L-glutamine (2 mM, Life Technologies; #A2916801) and incubated under standard tissue culture conditions (37˚C and 5% CO2). The oesophageal epithelial line EPC2-hTERT was grown in KSFM (Life Technologies; #17005075) supplemented with human recombinant epidermal growth factor (5g/L) and bovine pituitary extract (50 mg/L).

For subculture of OAC lines, cells were detached with trypsin (0.25%, 1 mL, 5 minutes, 37˚C), and neutralised with fresh growth media. For subculture of EPC2-hTERT, soy-bean

Historic Interpretation of N-Acetylcysteine and Cuproptosis

New Interpretation of N-Acetylcysteine and Cuproptosis

**Fig 1. Schematic of historic and proposed N-acetylcysteine mechanism in the context of Cuproptosis.**

trypsin inhibitor (250 mg/L, 5 mL) was added to neutralise the trypsin and centrifuged (5 minutes, 250 x g). The cell pellet was then re-suspended in fresh media.

Cells were seeded (50 μL per well) at 1500 cells per well except SK-GT-4 which was seeded at 1000 cells per well into 384-well, CELLSTAR® Cell Culture Microplates (Greiner, #781091), and incubated under standard tissue culture conditions for 24 hrs before the addition of compounds.

## Cell survival assays

**Copper Ionophore and N-acetylcysteine dose responses.** Cells were plated as described above. Disulfiram and elesclomol dose responses were carried in the presence and absence of a 15 min pre-incubation with 1 or 10 mM NAC (A7250; Sigma). NAC was made up in media and 0.1M NaOH was used to neutralise the media and bring the pH back to 7.4. This was then syringe filtered before being added to the cells. Elesclomol and Disulfiram were added and incubated for 48hrs before plates were fixed and stained with Hoechst 33342 (#H1399; Mol. Probes) and imaged on the ImageXpress micro XLS (Molecular Devices, USA). Images were analysed on the MetaXpress software and dose responses were assessed using the total nuclei counts per well normalized to DMSO controls to calculate percentage cell survival.

**$H_2O_2$ dose response.** Cells were plated as described above. $H_2O_2$ (H1009; Sigma) was diluted in media to give an 18 point dose response starting at 1 mM and incubated with the cells for 48 hrs. Cell viability was assessed by alamar Blue assay [26].

### Inductively coupled mass spectrometry

$5 \times 10^6$ cells were seeded in a T175 flask and incubated in 25ml RPMI overnight. Media was then replaced with the addition of compound treatments (DMSO (0.1%), disulfiram (600 nM), or elesclomol (200 nM)) with or without a 15 min pre-incubation with 1mM or 10 mM NAC, before further incubation for 6 hrs when cells were then collected. Media was removed and cells were washed in PBS, typsinised and counted. For each sample $1 \times 106$ cells were pelleted in an Eppendorf, the supernatant was discarded and samples were frozen at -80˚C.

For analysis, samples were thawed and concentrated nitric acid was added (100 μL per sample) and mixed. Samples were then vortexed and sonicated and left overnight at room temperature. Samples were made up to 1 mL using Di water and then further diluted tenfold and analysed in the ICP Facility, School of Chemistry, University of Edinburgh.

### Nanostring transcriptomic analysis

All NanoString nCounter analyses were carried out on the Human PanCancer Pathways (Catalogue number XT-CSO-PATH1-1) and Metabolic Pathways (Catalogue number XT-CSO-HMP-12) panels, covering 1449 genes.

Cells were seeded at $8 \times 10^4$ cells in 6-well plates and incubated under standard tissue culture conditions for 24 hrs. For basal analysis media was then removed and plates were washed twice with ice cold PBS before being snap frozen at -80. For treatment induced analysis, media was replaced with fresh media containing compound treatments (DMSO (0.1%) or elesclomol (200 nM)) before further incubation for 6 hrs, and then samples were processed the same as for basal samples.

For RNA extraction cells were scraped and lysed using QIAshredders (#79654, QIAGEN), RNA was extracted by means of the Qiagen RNeasy Mini kit (#74104, QIAGEN) (with β-mercaptoethanol) according to manufacturer instructions, and included a DNase digestion step (#79254, QIAGEN).

## Results

We have explored NACs ability to alleviate the toxicity of the copper ionophores elesclomol and disulfiram, in two oesophageal cancer lines (OAC-P4C and SK-GT-4). We found, in line with previous research [10, 13], that pretreatment with NAC alleviated elesclomol and disulfiram induced cell death (Fig 2).

However, while 10mM NAC is sufficient to relieve Disulfiram and Elesclomol induced toxicity we found that 1mM NAC does not relieve Elesclomol induced toxicity at all, which has been observed in other cell lines previously [13], but is capable of partially rescuing Disulfiram induced toxicity. The high concentrations required to alleviate cytotoxicity suggest that in this context NAC may be working via mechanisms that are independent or in addition to ROS antagonism for example it has been shown that even submillimolar concentrations of NAC significantly stimulate GSH synthesis [20, 27]. This observation is further supported by results demonstrating other antioxidants, for example trolox [28], and Coenzyme Q10 [13] do not alleviate the activity of elesclomol or disulfiram.

Given these findings and the dual nature of NAC as an antioxidant and the less well-known action as a copper interactor, we felt it was important to assess NACs effect on the intracellular copper levels since it could be acting via either mechanism. To our knowledge this has not been assessed previously. To quantify this, we measured the intracellular copper levels after Elesclomol and Disulfiram treatment with and without NAC preincubation at two concentrations using inductively coupled plasma mass spectrometry (ICP-MS) in the two most sensitive cell lines; OAC-P4C and SK-GT-4. Importantly we demonstrate here for the first time that

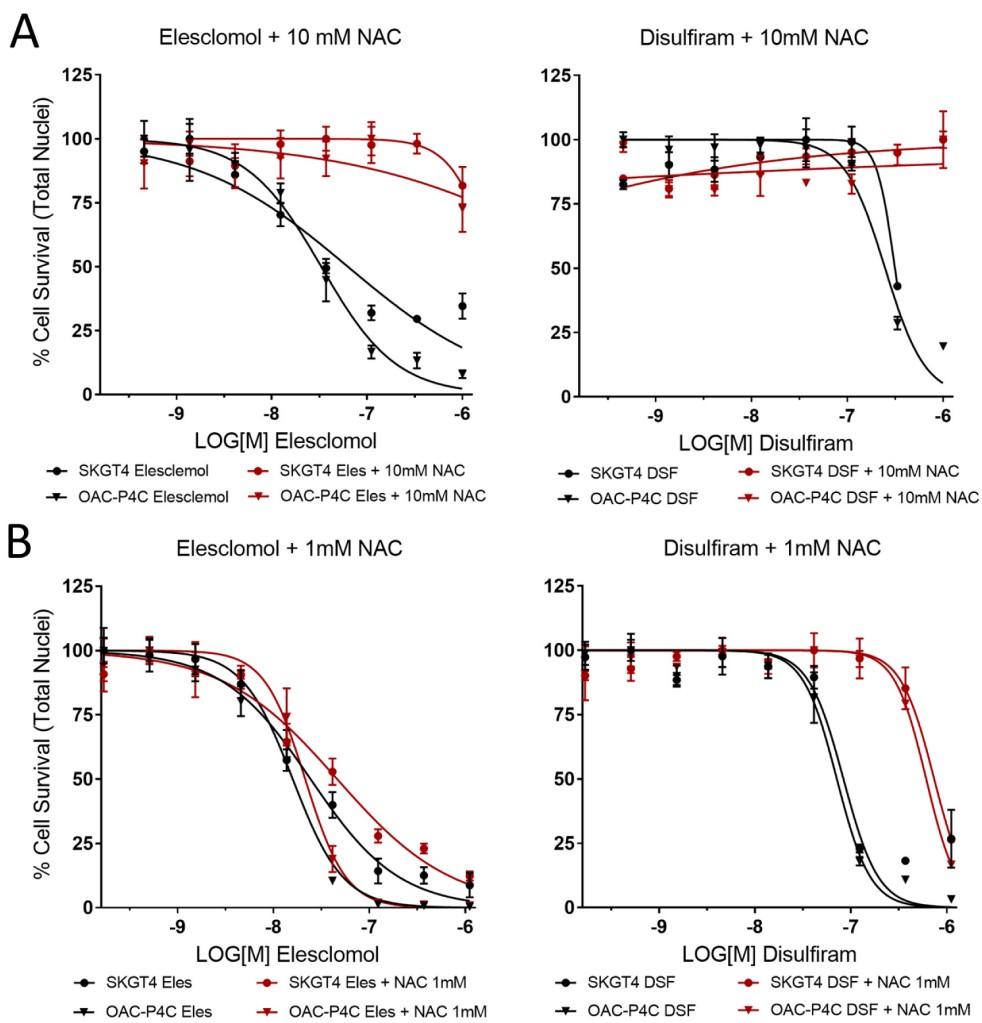

**Fig 2. N-acetylcysteine treatment and copper ionophore induced cell death.** Elesclomol and Disulfiram dose response in OAC-P4C and SK-GT-4 cells with A) 10mM N-acetylcysteine, B) 1mM N-acetylcysteine. NAC = N-acetylcysteine.

pre-incubation of cancer cells with NAC leads to loss of copper accumulation post copper ionophore treatment, and results in copper levels comparable to those of the untreated cells (Fig 3). Importantly the copper levels reflect the cytotoxicity data, where only 10mM NAC is able to reverse the cytotoxicity and copper accumulation associated with Elesclomol, while both 1 and 10mM NAC are able to reverse the cytotoxicity and copper accumulation associated with Disulfiram. These results indicate that NAC pretreatment dramatically prevents cellular accumulation of copper via the copper ionophores. Therefore, NAC pre-incubation appears to prevent compound activity by preventing the increase in intracellular copper levels and not via antioxidant properties.

To begin to investigate the mechanism of copper mediated cell death and any role of ROS, we explored elesclomol induced transcriptomic changes across a panel of oesophageal cancer cell lines using the NanoString nCounter platform S1 Table. Firstly, assessment of the enzymes involved in maintenance of redox homeostasis did not reveal any major induction of antioxidant enzymes, including catalase (CAT) and superoxide dismutase (SOD) (Fig 4), suggesting

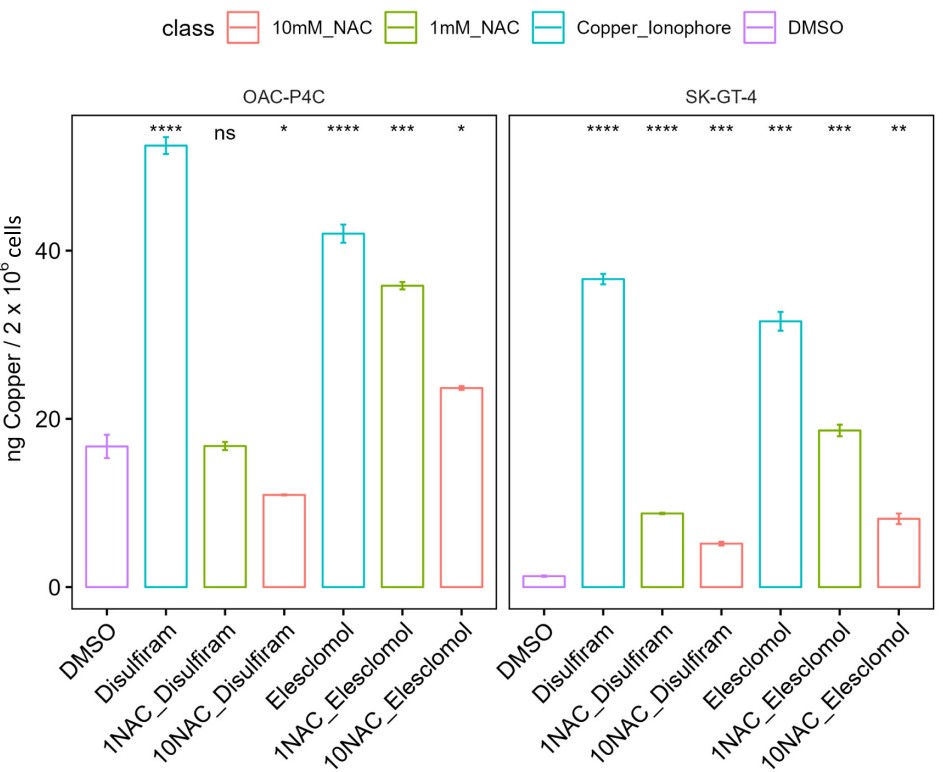

**Fig 3. NAC prevents the intracellular increase in copper caused by copper ionophores.** ICP-MS intracellular copper levels in the cell lines OAC-P4C and SK-GT-4 after Elesclomol (200nM) or Disulfiram (600nM) treatment with and without 15 min, NAC pre-treatment at 1 or 10mM. NAC = N-acetylcysteine, ICP-MS = inductively coupled masspectrometry. N = 3. P-values (Student's T-test) compared to DMSO treatment for each cell line. * = p<0.05, ** = p<0.01, *** p<0.005, **** p<0.001.

the cells do not mount an antioxidant response. However, elesclomol treatment does lead to an increase in two genes involved in thiol homeostasis, TXNRD1 and GCLC, though this may be due to direct interaction of copper with thiol groups [29, 30] rather than a result of ROS production.

Furthermore, we compared cell sensitivity to hydrogen peroxide ($H_2O_2$). If ROS levels are responsible for the cell death associated with the copper ionophores then we would expect the differential sensitivity observed across oesophageal cell lines to disulfiram and elesclomol induced cytotoxicity to correlate with $H_2O_2$ sensitivity. We have previously profiled the sensitivity of 10 oesophageal cancer cell lines and two tissue matched controls to elesclomol and disulfiram and shown that the correlation in sensitivity to the two drugs is very high (0.94) [1]. Here we have taken the most sensitive cell line OAC-P4C and the most resistant cell line FLO-1, and compared their sensitivity to $H_2O_2$. Results show that the copper ionophore resistant cells, FLO-1, are significantly more sensitive to oxidative stress via $H_2O_2$ than the copper ionophore sensitive cells, OAC-P4C, or tissue matched control cells (EPC2-hTERT) (Fig 5). This is consistent with previous findings that FLO-1 have a lower antioxidant capacity and increased sensitivity to ROS inducing compounds [31]. Overall, these findings do not fit with the hypothesis that ROS levels are responsible for copper mediated cell death, particularly in relation to the copper ionophores elesclomol and disulfiram.

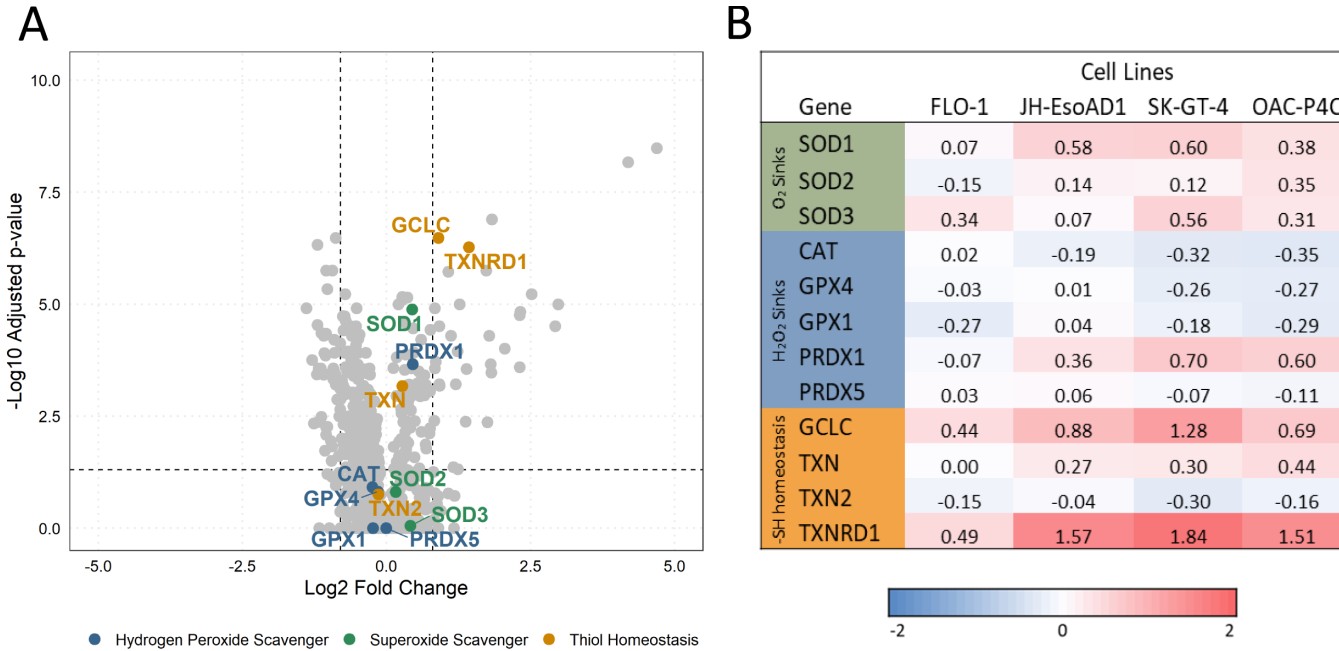

**Fig 4. Elesclomol induced transcriptomic changes in redox homeostasis genes.** A) Volcano plot for elesclomol induced differential expression analysis across FLO-1, JH-EsoAD1, SK-GT-4 and OAC-P4C cells. Redox homeostasis genes hydrogen peroxide scavengers (Blue), Superoxide scavengers (Green), Thiol homeostasis genes (Mustard) are labelled. B) Elesclomol induced transcriptomic log2 fold changes for highlighted redox homeostasis enzymes across four separate cell lines; FLO-1, JH-EsoAD1, SK-GT-4 and OAC-P4C cells.

## Discussion

A new form of cell death termed cuproptosis has been discovered and reviewed extensively by others [32–34]. This new form of cell death has been widely studied in relation to a novel class of copper ionophores, including elesclomol and disulfiram. While there is a wealth of evidence that these compounds lead to the intracellular accumulation of copper, the exact mechanism leading to cell death remains contentious.

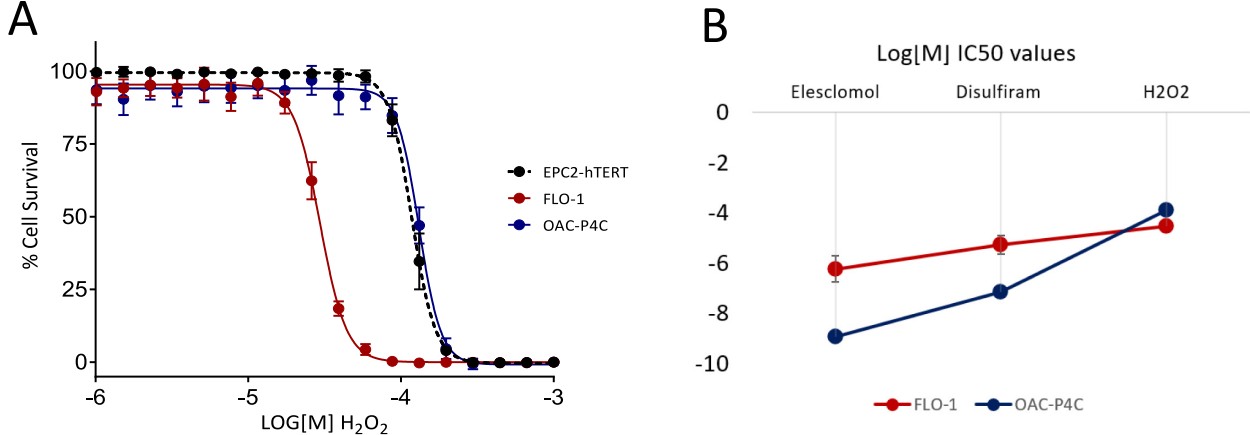

**Fig 5. Sensitivity to copper ionophores and H₂O₂.** A) $H_2O_2$ dose responses for OAC cell lines OAC-P4C and FLO-1, and tissue matched control EPC2-hTERT. B) Log molar IC50 values for Elesclomol, Disulfiram and $H_2O_2$, across the FLO-1 and OAC-P4C cell lines.

It has been proposed that the intracellular copper accumulation leads to the production of ROS and that this is the cause of cell death [4, 10–13]. Much of the evidence for this is based on the ability of the small molecule NAC to alleviate the toxicity of the drugs [10, 12, 13, 19]. Here we have demonstrated across multiple cell lines and drugs that while NAC does indeed alleviate the cytotoxic effects of these copper ionophores it is only at concentrations where it significantly reduces copper cellular uptake. This suggests that it is the reduction in copper accumulation that is responsible for the diminished cell death. Given that NAC has and is still widely used in cell culture-based studies investigating the role of ROS generation in copper toxicity, we believe it is important to disseminate the finding that NAC interferes with the intracellular accumulation of copper caused by copper ionophores, confounding conclusions drawn as to the effects of ROS.

We do not know the exact mechanism through which NAC significantly reduces copper cellular accumulation caused by the ionophores but disulfiram requires Cu(II) or an acidic environment for its conversion to an active metabolite diethylthiocarbamate (DETC) [5], and since NAC is a potent reducing agent and has been shown to potently reduce Cu(II) to Cu(I) [35] this is likely to prevent the conversion of Disulfiram to DETC, blocking its ability to act as a copper ionophore. Elesclomol on the other hand does not require conversion for its activity, however Elesclomol chelates Cu(II) [4] so if the copper in the culture media is reduced by NAC then this is likely to prevent the copper ionophore effect of Elesclomol. This may also explain why only high concentrations of NAC are able to block elesclomol activity. It is also possible that NAC is directly interacting with the copper or elesclomol itself, leading to sequestration [20, 24].

Our results also demonstrate a lack of evidence that ROS is the major contributor to cuproptosis and copper ionophore induced cell death. Our transcriptomic results suggest that cells do not mount a strong antioxidant reaction in response to the copper ionophores. In fact, $H_2O_2$ sink enzymes glutathione peroxidase 1 and 4 (GPX1, GPX4), and catalase (CAT) are marginally downregulated. These copper ionophore mediated transcriptional changes mirror almost perfectly the results of a similar study, using a different cell model system, on cuproptosis using excess concentrations of Cu (II) in the culture media [35] where they also demonstrate that there is no upregulation of antioxidant enzymes in response to excess copper.

While we see no induction of $O_2•^-$ or $H_2O_2$ cellular antioxidants in our cell lines, in future work it will be important to rigorously assess which if any ROS species are produced by elesclomol and disufiram. Further it will be important to assess whether specific scavengers are able to rescue the cytotoxicity in order to conclusively evaluate whether ROS induction contributes to elesclomol and disulfiram induced cell death. Extensively testing the effects of specific antioxidants such as thiourea [36], ebselen [37], and trolox [38], may help to elucidate the contribution, if any, of ROS species towards elesclomol and disulfiram induced cell death (see [20] for a review of ROS scavengers and their properties). However, it is vital to ensure that these scavengers do not also affect intracellular copper accumulation, or directly interact with copper or the ionophores since for example ascorbic acid is a well-known scavenger of $O2•^-$ and $H_2O_2$ in vitro, but it is also known that in the presence of metals, such as iron, it becomes a powerful source of ROS [39]. Furthermore, Glutathione another well-known scavenger has been shown to prevent copper mediated toxicity but critically via direct binding and sequestering of copper ions [35], again confounding conclusions drawn from studies of copper mediated toxicity and the antioxidant glutathione.

Finally, emerging evidence indicates that copper induced cytotoxicity may be mediated via multiple mechanisms which are likely to be context dependent and different between the chemicals used to induce death (e.g. copper ionophores, interaction of agents with excess levels of endogenous or addition of exogenous copper, and other copper interacting compounds)

possibly explaining the conflicting evidence with regards to the mechanism of copper induced toxicity. For example, the anticancer activity of plant polyphenols has been hypothesized to involve mobilisation of endogenous copper and consequent DNA breakage [40, 41]. However, recent studies have shown that DNA damage was not induced in response to elesclomol treatment of colorectal cancer cells [12]. This suggests distinct mechanisms of copper mediated cell death, making it difficult to generalize across contexts.

Collectively our data indicate that NAC should be used with caution in experiments designed to ascertain the role of ROS in copper ionophore mediated cell death and cuproptosis in general and that this novel form of cell death requires further study to fully elucidate its mechanism.

## Supporting information

**S1 Table. Nanostring differential expression data.** Elesclomol 200nM vs DMSO. (CSV)

## Acknowledgments

We would like to thank A. Rustig, University of Pennsylvania for provision of the EPC2-hTERT cells and Lorna Eadies at the ICP Facility, School of Chemistry, University of Edinburgh for running the ICP-MS experiments.

## Author Contributions

**Conceptualization:** Rebecca E. Graham, Richard J. R. Elliott, Neil O. Carragher.

**Formal analysis:** Rebecca E. Graham, Richard J. R. Elliott.

**Funding acquisition:** Neil O. Carragher.

**Investigation:** Rebecca E. Graham, Richard J. R. Elliott, Alison F. Munro.

**Methodology:** Rebecca E. Graham.

**Project administration:** Neil O. Carragher.

**Supervision:** Neil O. Carragher.

**Visualization:** Rebecca E. Graham.

**Writing – original draft:** Rebecca E. Graham, Neil O. Carragher.

**Writing – review & editing:** Rebecca E. Graham, Richard J. R. Elliott, Alison F. Munro, Neil O. Carragher.

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
