## [Decision Letter · Decision Letter 0]

3 Sep 2023

PONE-D-23-23834A cautionary note on the use of N-acetylcysteine as a reactive oxygen species antagonist to assess copper mediated cell death.PLOS ONE

Dear Dr. Carragher,

Thank you for submitting your manuscript to PLOS ONE. After careful consideration, we feel that it has merit but does not fully meet PLOS ONE’s publication criteria as it currently stands. Therefore, we invite you to submit a revised version of the manuscript that addresses the points raised during the review process.

Please submit your revised manuscript by Oct 18 2023 11:59PM. If you will need more time than this to complete your revisions, please reply to this message or contact the journal office at plosone@plos.org. Please include the following items when submitting your revised manuscript:A rebuttal letter that responds to each point raised by the academic editor and reviewer(s). You should upload this letter as a separate file labeled 'Response to Reviewers'.A marked-up copy of your manuscript that highlights changes made to the original version. You should upload this as a separate file labeled 'Revised Manuscript with Track Changes'.An unmarked version of your revised paper without tracked changes. You should upload this as a separate file labeled 'Manuscript'.If applicable, we recommend that you deposit your laboratory protocols in protocols.io to enhance the reproducibility of your results. Protocols.io assigns your protocol its own identifier (DOI) so that it can be cited independently in the future. For instructions see: https://journals.plos.org/plosone/s/submission-guidelines#loc-laboratory-protocols. Additionally, PLOS ONE offers an option for publishing peer-reviewed Lab Protocol articles, which describe protocols hosted on protocols.io. Read more information on sharing protocols at https://plos.org/protocols?utm_medium=editorial-email&utm_source=authorletters&utm_campaign=protocols.

We look forward to receiving your revised manuscript.

Kind regards,

Abeer El Wakil, PhD

Academic Editor

PLOS ONE

Journal Requirements:

Additional Editor Comments:

The concept of the present study by Graham et al. entitled "A cautionary note on the use of N-acetylcysteine as a reactive oxygen species antagonist to assess copper mediated cell death" seems very interesting. The aim of the study is to investigate the recently identified mechanism of copper mediated cell death, cuproptosis. N-acetylcysteine has been conclusively shown to abrogate cuprotosis mediated via copper ionophores. Here we explore the mechanism through which N-acetylcysteine mediates relief of copper ionophore induced cell death and importantly demonstrate that it is not via the widely accepted effects of ROS antagonism but rather it interferes with the intracellular accumulation of copper mediated by copper ionophores. The concept is not yet fully explored, and it is very interesting to share such new observations that help other scientists in their studies.

In my opinion, the work provides an advance towards the current knowledge. The quality and content meet standards set forth by PLOS One journal and I suggest publishing the paper in the journal with minor changes and after addressing the points raised by the reviewers. Below are some suggestions that confirm reviewers' concerns, and it is important that the authors address them with attention:

- Keywords: Keywords enable authors to extend the representation of their manuscript content beyond that presented in the title and abstract. Keywords make your paper searchable and ensure that you get more citations. Thus, it is important to include the most relevant keywords that will help other authors find your paper. It is therefore very important to avoid repeating keywords already cited in the title to increase the discoverability of your paper. Please revise and address.

- The depiction of the research necessitates further elaboration. It is imperative to furnish additional details pertaining to the study both in the introduction and the discussion sections. Please elaborate and confirm with justified references.

- Please revise your references section and update it as much as possible. Some interesting recent references are missing.

Reviewers' comments:

Reviewer's Responses to Questions

**Comments to the Author**

1. Is the manuscript technically sound, and do the data support the conclusions?

Reviewer #1: Partly

Reviewer #2: Partly

2. Has the statistical analysis been performed appropriately and rigorously? 

Reviewer #1: No

Reviewer #2: Yes

3. Have the authors made all data underlying the findings in their manuscript fully available?

Reviewer #1: Yes

Reviewer #2: Yes

4. Is the manuscript presented in an intelligible fashion and written in standard English?

Reviewer #1: Yes

Reviewer #2: Yes

5. Review Comments to the Author

Reviewer #1: Major points:

1. Section Introduction: Surprisingly, the claim of the authors on line 59-60 is not supported by corresponding citations, although it is absolutely essential for this MS! Authors should pay careful attention to this part of the argument. For example, the references 4, 8, and 9 listed do not support your claim in any way!

2. Section Introduction: Likewise, the authors' claims on lines 60-64 are not even supported by relevant citations, or even true! There are many publications on the actual (not those who are mistakenly believed) properties of NAC, the most recent comprehensive article on the subject can be found here, for example: Mlejnek P. Antioxidants (Basel). 2022, 11(8): 1485. doi: 10.3390/antiox11081485.

3. Section Materials and Methods: Unfortunately, there is no description of the cell survival assay.

4. Section Results: The results presented in Figures 1 and 2 are trivial and, as the authors themselves state, are in agreement with the results of other authors. Therefore, it would be more appropriate to present them more economically, e.g. in the form of one table with the relevant IC50 values or only as one Figure!

5. Section Results: The assertion on lines 132-134 is not justified. In reference 13 you mentioned, the authors only speculate…

6. Section Results: In contrast, the key claim that NAC at high concentrations prevents the "uptake" of copper into cells is very poorly (!) supported experimentally:

a) It is shown only for elesclomol but not for disulfiram.

b) It is shown only for one cell line.

c) It is shown only for high NAC concentration.

d) There is no statistical evaluation of the results.

e) Figure 3 in this MS resembles Figure 4 from the work by Hughes et al., ACS Chem Biol. 2022, 17(7): 1876-1889. doi: 10.1021/acschembio.2c00301.

7. Section Results: It would be appropriate to show:

a) what ROS species are produced in your experimental system;

b) to find out whether their real scavengers, which do not simultaneously form chelates with copper, have an effect on survival or not. You can also find an overview of antioxidants and their properties in the above reference.

8. Section Results. Without the careful analysis suggested in point 7 (see above), the results presented in Figure 5 do not make much sense.

9. The Discussion section needs a very thorough overhaul.

Reviewer #2: This is an interesting study that cautions against using N-acetyl cysteine in studies on Cu mediated cell death. The manuscript can be accepted with minor revision.

1. The introduction needs more references to support their presumptions

2. The discussion section may benefit if the work of Hadi et al. on copper and polyphenol mediated cell death is also cited; in addition to other recent references.

3. Other ROS scavengers can perhaps also be discussed in some depth.

4. A schematic diagram will make the paper even easier to understand than it already is.

5. Can other statistical methods be used to compare and contrast results?

6. PLOS authors have the option to publish the peer review history of their article (what does this mean?). If published, this will include your full peer review and any attached files.

Reviewer #1: No

Reviewer #2: No

---

## [Author Response · Author response to Decision Letter 0]

17 Oct 2023

We have included a detailed point-by-point response to each of the reviewers comments in our uploaded response-to-reviewers document citing changes made to the revised manuscript. We have also responded to each of the editors points as outlined below:

- Keywords: Keywords enable authors to extend the representation of their manuscript content beyond that presented in the title and abstract. Keywords make your paper searchable and ensure that you get more citations. Thus, it is important to include the most relevant keywords that will help other authors find your paper. It is therefore very important to avoid repeating keywords already cited in the title to increase the discoverability of your paper. Please revise and address.

We thank the editor for the suggestion and have now added the following keywords:

Cuproptosis, Elesclomol, Disulfiram, N-acetylcysteine, Reactive Oxygen Species

- The depiction of the research necessitates further elaboration. It is imperative to furnish additional details pertaining to the study both in the introduction and the discussion sections. Please elaborate and confirm with justified references.

In line with the editor and reviewer comments we have now expanded and elaborated upon out text in the introduction and discussion to place our work into context of relevant published literature, citing additional references. We have also provided a new figure 1 to illustrate the main message of our study.

 - Please revise your references section and update it as much as possible. Some interesting recent references are missing.

We have now significantly revised our referenc.e section and updated with several new references (increasing from 21 to 41 cited references) including those recommended by both reviewers.

---

## [Decision Letter · Decision Letter 1]

30 Oct 2023

A cautionary note on the use of N-acetylcysteine as a reactive oxygen species antagonist to assess copper mediated cell death.

PONE-D-23-23834R1

Dear Dr. Carragher, 

We’re pleased to inform you that your manuscript has been judged scientifically suitable for publication and will be formally accepted for publication once it meets all outstanding technical requirements.

Kind regards,

Abeer El Wakil, PhD

Academic Editor

PLOS ONE

Additional Editor Comments (optional):

All comments have been nicely addressed with clear explanations. The article is suitable for publication in its current form.

Reviewers' comments:

Reviewer's Responses to Questions

**Comments to the Author**

1. If the authors have adequately addressed your comments raised in a previous round of review and you feel that this manuscript is now acceptable for publication, you may indicate that here to bypass the “Comments to the Author” section, enter your conflict of interest statement in the “Confidential to Editor” section, and submit your "Accept" recommendation.

Reviewer #1: All comments have been addressed

Reviewer #2: All comments have been addressed

2. Is the manuscript technically sound, and do the data support the conclusions?

Reviewer #1: Yes

Reviewer #2: Yes

3. Has the statistical analysis been performed appropriately and rigorously? 

Reviewer #1: Yes

Reviewer #2: Yes

4. Have the authors made all data underlying the findings in their manuscript fully available?

Reviewer #1: Yes

Reviewer #2: Yes

5. Is the manuscript presented in an intelligible fashion and written in standard English?

Reviewer #1: Yes

Reviewer #2: Yes

6. Review Comments to the Author

Reviewer #1: Authors have dealt with all my comments.

The article is now eligible for publication in the PLOS one.

Reviewer #2: The changes have been made in a nice manner with clear explanations. The article is suitable for acceptance in its current form.

7. PLOS authors have the option to publish the peer review history of their article (what does this mean?). If published, this will include your full peer review and any attached files.

Reviewer #1: No

Reviewer #2: No

---

## [Editor Report · Acceptance letter]

28 Nov 2023

PONE-D-23-23834R1 

A cautionary note on the use of N-acetylcysteine as a reactive oxygen species antagonist to assess copper mediated cell death. 

Dear Dr. Carragher:

I'm pleased to inform you that your manuscript has been deemed suitable for publication in PLOS ONE. Congratulations! Your manuscript is now with our production department. 

Kind regards, 

on behalf of

Professor Abeer El Wakil 

Academic Editor

PLOS ONE